# Extracellular Vesicles as Biomarkers in Chronic Hepatobiliary Diseases: An Overview of Their Interplay

**DOI:** 10.3390/ijms26136333

**Published:** 2025-06-30

**Authors:** Eleni Myrto Trifylli, Sotirios P. Fortis, Anastasios G. Kriebardis, Nikolaos Papadopoulos, Evangelos Koustas, Panagiotis Sarantis, Spilios Manolakopoulos, Melanie Deutsch

**Affiliations:** 1GI-Liver Unit, 2nd Department of Internal Medicine, National and Kapodistrian University of Athens, General Hospital of Athens “Hippocratio”, 114 Vas Sofias, 11527 Athens, Greece; trif.lena@gmail.com (E.M.T.); smanolak@med.uoa.gr (S.M.); meladeut@gmail.com (M.D.); 2Laboratory of Reliability and Quality Control in Laboratory Hematology (HemQcR), Department of Biomedical Sciences, Section of Medical Laboratories, School of Health & Caring Sciences, University of West Attica (UniWA), Ag. Spyridonos Str., 12243 Egaleo, Greece; sfortis@uniwa.gr; 3Second Department of Internal MedicinWe, 401 General Military Hospital, 11527 Athens, Greece; nipapmed@gmail.com; 4Oncology Department, General Hospital Evangelismos, 10676 Athens, Greece; vangkoustas@gmail.com; 5Department of Biological Chemistry, Medical School, National and Kapodistrian University of Athens, 11527 Athens, Greece; panayotissarantis@gmail.com

**Keywords:** exosomes, biomarker, extracellular vesicles, metabolic dysfunction-associated steatotic liver disease, cholangiocarcinoma, autoimmune hepatitis, hepatitis, hepatocellular carcinoma, cirrhosis, alcohol-related liver disease, cholangiopathies

## Abstract

Hepatobiliary diseases, which include disorders of the liver, gallbladder, and bile ducts, remain a major global health concern. A significant proportion of deaths worldwide are attributed to hepatic diseases, accounting for 4% of the total global mortality in 2023. Among benign hepatobiliary diseases, metabolic dysfunction-associated steatotic liver disease is the most prevalent liver pathology, with a concerning rise in incidence, while it is recognized as the leading cause of liver transplantation in the United States. However, there is a notable rise over time in cases of autoimmune hepatobiliary disorders, including autoimmune hepatitis, primary biliary cholangitis, and primary sclerosing cholangitis. Meanwhile, hepatocellular carcinoma still remains the most frequently diagnosed hepatobiliary malignancy, constituting the third leading cause of malignancy-related mortality globally. Meanwhile, cholangiocarcinoma and gallbladder cancer are the second and third most common hepatobiliary malignancies, respectively, both exhibiting highly aggressive malignant behavior. Despite the notable advances in biomarkers and the development of therapeutic tools, early diagnosis and monitoring are considered pivotal for the management of the aforementioned pathologies. The development of new non-invasive biomarkers that can effectively identify, monitor these pathologies, and guide their management is considered a necessity. Extracellular vesicles (EVs) constitute nanoparticles with several embedded cargoes, with a significant role in intercellular communication, which are considered promising biomarkers in several diseases, including viral, metabolic, autoimmune, and malignant diseases. In this review, we will shed light on the role of EVs as novel frontiers in hepatobiliary diseases.

## 1. Introduction

Hepatic disorders and cholangiopathies comprise a wide range of diseases affecting hepatocytes, the biliary tree, and liver vasculature, posing a significant global health burden. Hepatic disorders involve metabolic, structural, and functional impairments and are classified accordingly. Some of the most common metabolic hepatic disorders are metabolic dysfunction-associated steatotic liver disease (MASLD), alcohol-associated liver disease (ALD), and their combination, the so-called metabolic-alcohol-associated liver disease, while viral hepatitis B and C constitute hepatic diseases that have significantly altered global mortality and morbidity for decades [1]. Meanwhile, some of the most commonly diagnosed autoimmune hepatobiliary disorders are autoimmune hepatitis (AIH), primary biliary cholangitis (PBC), and primary sclerosing cholangitis (PSC) [1]. All the above disorders may eventually lead to several liver-related outcomes such as fibrogenesis, cirrhosis, as well as hepatobiliary oncogenesis, including gallbladder cancer (GC), hepatocellular carcinoma (HCC), and cholangiocarcinoma (CCA). HCC constitutes the most frequently diagnosed primary liver malignancy and the third cause of cancer-related death [2], while CCA is the second most diagnosed malignancy, followed by GC, which both constitute highly malignant diseases [2,3].

Despite advancements in diagnostic and therapeutic modalities, these conditions continue to impact many individuals globally, and the management of the aforementioned hepatobiliary diseases remains challenging. Extracellular vesicles (EVs) have been in the spotlight of biomarker research. These nanoparticles have a key role in intercellular communication, as they carry several types of cargo, which can alter the functional state of the recipient cells. The type of cargo can influence disease development and progression, and it can potentially serve as a target for the identification of EVs related to the disease [4]. In this review, we will shed light on the current knowledge regarding the emerging role of EVs as novel frontiers in the most commonly diagnosed benign and malignant hepatobiliary diseases, and we will provide a concise overview of their interplay.

## 2. Review of EV Biogenesis

EVs constitute quite heterogeneous, nanosized particles that are distinctly classified into 3 subclasses, which are closely related to their biogenesis pathway (Figure 1). More particularly, the 3 subclasses are (i) exosomes, (ii) microvesicles, and (iii) apoptotic bodies, starting with the smallest diameter (40–150 nm), towards the intermediate (150–1000 nm), and the largest (>1000 nm), respectively. There are three discrete pathogenetic pathways, including (a) inward membrane budding for exosome biogenesis, (b) outward membrane blebbing for microvesicles’ production, and (c) cellular apoptotic mechanism, for the biogenesis of apoptotic vesicles and eventually of apoptotic bodies [4,5]. However, their heterogeneity is also identified in the wide variety of cargoes that they can encompass. There is a multitude of cargoes, including coding and non-coding RNAs, autophagosomes, DNA molecules, several receptors, and other protein and lipid molecules, which can significantly alter the function of several other cells via intercellular communication. More specifically, the parental cell that produces several types of EVs, regarding their size and cargoes, can significantly alter the functional state of the recipient cells, which can receive these particles through several pathways. These nanoparticles are delivered either via receptor–ligand interaction or through a wide variety of endocytic processes, including receptor, lipid, or caveolin-mediated endocytosis, as well as micropinocytosis. Nevertheless, the delivery of the cargoes and their interactions with the recipient cells constitute mechanisms that are still under study [4,5,6].

Exosomes are formed through inward budding of the plasma membrane, involving endosomal pathways. Early endosomes mature into late endosomes, which form intraluminal vesicles (ILVs) through inward invagination. These are then packaged into multivesicular bodies (MVBs). The Endosomal Sorting Complex Required for Transport (ESCRT) 0-III complex guides this process. ESCRT 0 and I initiate vesicle formation, and ESCRT-II recruits ESCRT-III to finalize ILV detachment. MVBs may fuse with lysosomes for degradation or with the plasma membrane to release exosomes [8]. Exosome formation may also occur via ESCRT-independent routes involving the membrane or cytoplasm. Fusion with autophagosomes creates amphisomes, which may also release exosomes. SNARE proteins (like Ykt6, VAMP7, and syntaxin 1A) mediate exosome exocytosis [8]. Microvesicles (150–1000 nm) bud directly from the cell membrane. Under hypoxic conditions, SNAREs and Rab-GTPases are involved in cargo recruitment. Under normoxia, ARRDC1 and TSG101 regulate this process. Additionally, ARF6 facilitates the packaging of MHC-I, RNAs, DNAs, and integrins [9]. Moreover, apoptotic bodies, the largest EVs, are released during programmed cell death. Apoptosis leads to nuclear fragmentation, chromatin condensation, and disintegration of organelles. Membrane protrusions like apoptopodia and microtubule spikes eventually form apoptotic vesicles, which segment into apoptotic bodies [10].

## 3. An Overview of the Interplay Between EV and Hepatobiliary Diseases and Their Potential Role as Biomarkers

### 3.1. Chronic Viral Hepatitis B and C

EVs are key in cell-to-cell communication between viral hepatitis B or C (HBV or HCV) and host cells, facilitating disease progression and chronicity. After six months, this leads to liver tissue destruction, fibrotic injury, cirrhosis, and hepatocellular carcinoma (HCC) development [7]. Infected host cells secrete EVs that contain viral elements, while these EVs are quite similar to viruses, as they share biogenesis, uptake mechanisms, and secretory pathways. Notably, viral nanoparticles (30–1000 nm) and EVs resemble each other in size, morphological heterogeneity, and transported cargoes. HBV and HCV are common chronic hepatitis viruses that are about 42 and 50 nm, respectively. Like EVs, viral particles are double-membrane vesicles carrying DNA (HBV) or RNA (HCV). Both utilize ESCRT-dependent and independent biogenesis pathways. Viruses can manipulate EV biogenesis to enhance transmissibility and contagiousness (e.g., HBV and HCV) [11,12,13,14].

EVs containing HBV DNA isolated from chronic HBV patient serum/plasma can infect non-infected HepG2 hepatocytes, confirmed by positive HBcAg, HBsAg, and DNA levels. Similarly, in mice, EVs from infected hepatocytes mediate HBV transmission to healthy ones [11,12]. Chronic HBV patients have elevated circulating EVs with viral genetic material and proteins that impair immune responses, notably reducing NK cell cytotoxicity and interferon gamma (INF-γ) production [11,12,13,14]. HBV-induced immune suppression is also mediated via EV-contained microRNAs (e.g., miR-21), which inhibit interleukins like interleukin (IL)-12, weakening the immune response and promoting HCC development [15]. Infected hepatocytes secrete exosomes and virions that overregulate programmed death-ligand 1 (PD-L1) and downregulate CD69, suppressing leukocyte activity, promoting viral replication, and hindering HBV eradication in mouse models [11,12,13,16]. Additionally, exosomes that are derived from HBV-infected LO2 cells and contain miR-222 promote LX-2 cell activation in animal models (mice), promoting fibrogenesis [17]. Similarly, HCV-infected hepatocyte-derived EVs containing viral RNA infect non-infected hepatocytes, promoting replication. In chronic HCV, high EV-contained RNA levels may bypass oral antiviral treatment [18]. EV-contained miR-19a, abundant in chronic infection, activates HSCs and promotes fibrosis [19], while platelet-derived EVs are more abundant in chronic HCV than HBV, correlating with fibrotic markers (procollagen III N-terminal propeptide, hyaluronate), though further in-depth research is needed [20]. HCV-derived EVs also engage in cross-talk with monocytes (immune escape) and HSCs (fibrosis) [13]. In Table 1, we demonstrate some of the promising EV-based diagnostic biomarkers for chronic hepatitis B and C.

### 3.2. Metabolic Diseases

#### 3.2.1. MASLD

MASLD constitutes the major cause of chronic hepatopathies, with a big portion (20–30%) of the global population being affected, especially in the West. It includes a spectrum of pathological manifestations: steatosis (MASL), steatohepatitis (MASH), and cirrhosis. Among MASL patients, 3 of 10 will develop steatohepatitis, with or without fibrosis [23]. HCC can develop in cirrhosis and MASH alone, while MASLD is the leading cause of liver transplantation in the US. The scientific community is striving to develop non-invasive biomarkers, especially for early MASLD diagnosis [24]. Although liver biopsy is the gold standard for MASH and fibrosis diagnosis, it has limitations: invasiveness, complications, cost, sampling errors, inter-observer variability, inadequate sampling, and misinterpretation due to disease heterogeneity [25].

EVs are promising biomarkers for MASLD, playing a key role in its development and progression. They can be produced by hepatocytes or non-parenchymal liver cells like liver endothelial sinusoidal cells (LSECs), hepatic stellate cells (HSCs), portal fibroblasts, pre-adipocytes, and monocytes/macrophages [26]. EV alterations in quantity and quality are linked to MASLD pathogenesis and progression, especially in advanced cirrhosis (F3–F4) versus F1–F2 and healthy donors [27]. Patients with advanced fibrosis or cirrhosis show fewer circulating EVs, mainly from white blood cells and endothelial cells [28]. Similarly, in animal models of MASH and advanced fibrosis/cirrhosis, circulating EV levels are altered [29]. In mice, EVs from a rich fat diet injected into chow-fed mice increased inflammation, underscoring their role in MASH [30]. These vesicles can also identify MASLD among other chronic liver disorders, supporting their diagnostic and prognostic potential. Understanding MASLD progression, which is deeply rooted in cell-to-cell crosstalk, is essential for targeting EV cargos.

Lipotoxic hepatocytes, as parental EV producers, induce changes in parenchymal and non-parenchymal liver cells, such as hepatocytes, HSCs, macrophages, and LSECs, respectively, leading to inflammation, fibrogenesis, immune cell recruitment, and angiogenesis [31]. Lipid overload in the parenchyma triggers inflammation, which is amplified by monocytes/macrophages releasing IL-1b and IL-6. T-helper 17 cells, neutrophils, and Kupffer cells further activate HSCs via IL-17, IL-22, and TGF-β, respectively [32]. Quantitative and qualitative abnormalities in hepatocyte-derived EVs are reported in MASLD patients [33]. Moreover, HSC-derived EVs are also related to fibrogenesis and increased angiogenesis in the parenchyma. More particularly, it has been demonstrated that HSC-derived-EVs interact with LSECs, promoting angiogenesis and fibrosis via containing several protein molecules that promote fibrogenesis [34]. Meanwhile, it has to be underlined that angiogenesis is also promoted via Portal fibroblast-derived EVs that contain VEGF, a growth factor for endothelial cells [35]. Additionally, LSECs interact with HSCs via the release of LSEC-derived EV-sphingosine kinase 1 and/or EV-sphingosine-phosphate (S1P), a phenomenon that leads to HSCs activation, promoting fibrosis and MASLD progression [36]. These changes highlight the potential of EVs as a “liquid biopsy” for identifying disease and possibly stratifying severity [37]. Monocytes, macrophages, and platelets as parental cells also have a key role in EV-related MASLD development and disease progression [38]. On top of that, it has been demonstrated that there is a notable increase in circulating microvesicles that originate from monocytes, platelets, as well as endothelial cells in type 2 diabetes mellitus patients, which is one of the major aggravating factors for MASLD development and fibrosis progression [39]. Table 2a demonstrates several types of hepatocyte-derived EVs and their implication in MASLD progression (Table 2b).

#### 3.2.2. Alcohol-Associated Liver Disease (ALD)

Excessive alcohol consumption is closely associated with liver toxicity, including steatosis, steatohepatitis, and cirrhosis [64]. ALD shows aberrations in EV features, including increased circulating EV levels compared to healthy controls, as shown in animal ALD (mice) models [65]. Most circulating EVs originate from hepatocytes and HSCs, with levels proportionally increasing with disease severity, such as in alcoholic hepatitis (AH) compared to alcohol-abusing patients without AH [66]. These EVs significantly alter recipient cells like HSCs, macrophages, and endothelial cells [66,67]. In AH, hepatocytes release EVs that increase macrophage IL-17 and IL-1β expression, promoting fibrosis. HSCs are also activated by hepatocyte-derived EVs containing collagen a-SMA, demonstrated in AH models (mice) [66,67,68]. Additionally, hepatocyte-derived EV-CD40 ligand stimulates macrophages, while mitochondrial double-stranded RNA EVs induce neutrophilic infiltration, interacting with Kupffer cells and triggering IL-1b release [69].

ALD-induced EV miRs also promote inflammation and fibrotic injury. HSC-derived EVs-miR-19b play a key role in HSC activation and fibrogenesis [66,67]. AH-injured hepatocytes release EV-miR-181 and EV-miR-27a, promoting fibrosis via HSC activation in mice [69]. Monocyte-derived EV-miR-27a promotes M2 phenotype polarization of monocytes [70]. Macrophage activation (M2 phenotype) is also induced by hepatocyte-derived EVs containing heat shock HSP90 protein, shown in ALD animal models, while hepatocyte-EV CYP2E1 induces alcohol-related monocyte toxicity [67]. There is an interplay between EV biogenesis and the autophagy pathway. EVs suppress autophagy via miR-155 overexpression, decreasing LAMP1 and LAMP2 levels [71]. Thus, EVs with miR-155 inhibit autophagy in hepatocytes, a key toxic molecule-recycling process [71]. Hepatocyte-originated EVs containing miR-122 play a critical role in inflammation, inducing lipopolysaccharide and pro-inflammatory cytokine stimulation in monocytes [34]. Table 3 demonstrates EV-based biomarkers specifically associated with ALD and AH.

#### 3.2.3. Hereditary Hemochromatosis (HH)

HH is a homeostatic disorder of iron, caused by a high iron Fe gene mutation (C282Y SNP), leading to decreased hepcidin despite increased iron, as well as increased intestinal absorption of it [74]. The iron overload in hepatic parenchyma leads to liver fibrosis, cirrhosis, and an increased risk of HCC [75]. EVs have a key role in iron homeostasis, as they carry ferritin and transferrin and can modulate hepcidin function and ferroptosis [76]. Under oxidative stress and iron overload, cells release more EVs containing antioxidant protein molecules and iron metabolism proteins, which help minimize injury and modulate ferroptosis [76]. Iron-embedded EVs can prevent injury in parental cells by redistributing iron but may damage recipient cells, contributing to disease progression in hepatic disorders like MASLD and HH [77,78]. Hepatocyte- and macrophage-derived EVs under iron-rich conditions show altered ferritin and iron-handling enzymes, serving as sensitive biomarkers of intracellular iron status [79]. Given their presence in fluids like blood, urine, and bile, EVs offer a non-invasive method to monitor iron accumulation and liver damage in HH. In Table 4, we summarize the key points of iron uptake regulation and EVs’ role in hemochromatosis diagnosis.

### 3.3. Autoimmune Hepatobiliary Diseases

#### 3.3.1. PBC and PSC

The emerging role of EVs in biliary tract pathophysiology is highlighted in current studies. EVs and their embedded cargoes are involved in the physiological function of cholangiocytes and cholangiopathies such as PBC and PSC [80]. Released from the apical part of polarized cholangiocytes, EVs contribute to biliary tract homeostasis by mediating cross-talk between parental cells and recipient cells like endothelial cells and hepatocytes, while hepatocyte-originated EVs carrying epidermal growth factor receptor (EGFR) and integrin beta 4 (ITGB4) are implicated in biliary tract oncogenesis [81,82]. PBC and PSC are characterized by cholestasis, inflammation, and fibrogenesis, with ductular reaction and cholangiocyte activation. EVs are closely related to cholangiocyte proliferation, where bile-originated EVs interact with primary cilia to suppress ERK signaling through miR-15a overexpression [83]. EVs from biliary tract cells also promote fibrotic injury, particularly those carrying long non-coding RNA (lncRNA) H19. These vesicles mediate interaction between cholangiocytes and hepatocytes, contributing to fibrosis progression in PSC patients. EV-H19 increases high mobility group AT-hook 2 levels, stimulating macrophage and HSC activation and chemotactic pathways [84].

#### 3.3.2. Autoimmune Hepatitis (AIH)

AIH is an immune-mediated liver disease that can occur at any age and may lead to cirrhosis and transplantation [83]. Though the pathogenetic mechanism remains unclear, AIH is linked to auto-antibody production (antinuclear antibody, liver kidney microsomal type 1 and 3 antibody; anti-smooth muscle antibody, anti-liver-cytosol-1), cytokine overproduction (TNF-α, IL-1, -6, -17, -12, INF-γ), and T-cell imbalances [85]. Concanavalin A and hepatic S100 proteins induce AIH and serve as targets for mesenchymal stem cell (MSC)-derived EVs [86]. These EVs alter the AIH microenvironment, reducing inflammation and fibrosis. MSC-exosomes-miR-223 show liver-protective, anti-inflammatory, and anti-apoptotic effects by stimulating signal transducer and activator of transcription 3 (STAT3) in macrophages and hepatocytes [2]. EVs with miR-21 and miR-16 promote pro-inflammatory macrophages. T cells show functional aberrations in AIH. Regulatory T cell, T helper (Th) cell type-17, -1, -2 imbalances are addressed by MSC-EV-sphingosine 1-phosphate, which downregulates Th-17. MSC-exosomes promote Th-1 to Th-2 transition and inhibit T-cell immune function via PD-L1 overexpression [85]. Table 5 presents the role of EVs as biomarkers for autoimmune hepatobiliary diseases.

### 3.4. Complications of Chronic Hepatobiliary Diseases

Chronic hepatobiliary diseases may gradually lead to the development of liver-related outcomes and hepato- or cholangiocarcinogenesis. In this section, we focus on fibrogenesis, cirrhosis, biliary tract stenosis, as well as portal hypertension, ascites, hepatopulmonary and porto-pulmonary syndromes, coagulation disorders, and hepatobiliary malignancies. Meanwhile, chronic cholangiopathies might lead to biliary tract stenosis, chronic cholecystitis, and lithiasis, as well as GC [2].

### 3.5. Fibrotic Injury and Cirrhosis

The development of fibrotic injury and cirrhosis encompasses conditions that are characterized by increased fibrogenesis in the extracellular matrix, with the latter presenting distortion of hepatic tissue architecture and angiogenesis. Pro-fibrotic EVs have a key role in the activation of quiescent HSCs, such as EV-PDGF receptor α (PDGFRα), with the degradation of PDGFRα and MVBs being suppressed [50]. It has been demonstrated that patients with established cirrhosis have an increased amount of these EVs in their circulation compared with healthy individuals. Meanwhile, some other aberrations in EV levels that are identified under these fibrogenic conditions include the notably decreased amount of HSC-derived EVs that contain the Twist family basic helix-loop-helix transcription factor 1 (Twist1), as well as miR-214, leading to the paracrine activation of other HSCs [87]. Similarly, LSECs release EVs that contain SK1 in mice fibrosis models [88]. In addition, hepatocyte-originated EVs-miR-128-3p that are released under lipotoxic conditions interact with HSCs and subsequently activate them, promoting fibrogenesis through suppressing peroxisome proliferator-activated receptor (PPAR) γ, which has a pivotal role in quiescent HSC regulation and fibrosis inhibition in liver parenchyma, while the hepatocyte-derived EVs that carry miR-192 act in the same manner for HSC activation and fibrogenesis progression [89].

### 3.6. Portal Hypertension and Hepatopulmonary and Porto-Pulmonary Syndromes

Parenchymal structural modifications, due to fibrosis, lead to the establishment of portal hypertension (PT), hyperdynamic circulation, and portosystemic collaterals. Large-diameter EVs of unknown origin are implicated in systemic vasodilation, which has been found in patients with cirrhosis with a B or C grade of Child–Pugh (CP) score in comparison to healthy controls and CP A grade [90]. On the other hand, small EVs have been correlated with elevated intrahepatic resistance via over-activation of signaling pathways such as JAK2 and ROCK ones [7]. These EVs are received by endothelial cells, resulting in vasoconstriction suppression. EV-miR-194 levels are increased in hepatopulmonary syndrome in animal models, as well as portal myofibroblasts-EV-VEGF that is received by endothelial cells, leading to disease aggravation [91]. Meanwhile, EVs’ implication in porto-pulmonary hypertension has also been studied in animal models (mice), with its development via injection of EVs from diseased to healthy mice [92].

### 3.7. Coagulation Disorders

Coagulation disorders are closely associated with EVs-phospholipids and platelet-derived EVs-annexin V, with the latter being related to increased severity and mortality for cirrhotic patients [93]. Similarly, EV-Tissue factor is also implicated in coagulation and disease progression in animal and human cirrhotic models. Furthermore, increased angiogenesis is attributed to activated HSCs, cholangiocytes, and hepatocyte-originated EVs that are interacting with LSECs and lead to increased angiogenesis. Furthermore, increased angiogenesis is attributed to activated HSCs, cholangiocytes, and hepatocyte-originated EVs (e.g., EV-vanin-1 in mice models) that are interacting with LSECs and lead to increased angiogenesis, as well as endothelial cell-derived EVs (e.g., VEGF) that contain VEGF [19,26,49].

### 3.8. Ascites and Hepatic Encephalopathy

There are differences in the protein cargoes of circulating EVs between healthy controls and animal models of hepatic encephalopathy, while patients who present ascites have a notable increase in the amount of hepatocyte and endothelial-derived EVs. The former population of EVs is significantly associated with mortality, while small-sized vesicles inside the ascitic fluid have been correlated with patient proneness to inflammation [7,94].

### 3.9. Biliary Tract Stenosis and Chronic Cholecystitis

The cells from these stenotic areas produce a noticeable amount of EVs that enter the bile. The non-invasive identification of malignant common bile duct (CBD) stenosis, which can be present in chronic liver diseases such as PSC, is in the spotlight of the scientific community. Based on the study by Severino V et al., the concentration levels of bile-EVs could accurately (100% accuracy) identify the malignant CBD stenosis from the benign ones [95]. Dysbiosis constitutes a pivotal aggravating factor for cholecystitis development, with EVs playing a key role as they carry several bacteria, which are produced by microbial-infected cells. There are several qualitative aberrations in the serum exosomes between the chronic versus acute cases. Exosomes are significantly released under the condition of cholecystitis, a phenomenon that is closely related to exosomal function as regulators of gene expression and signaling pathways, which are implicated in cholecystitis. Interestingly, the bacterial and viral composition of the microbiome in the biliary tract can significantly affect the origin (e.g., host-infected cells) and cargoes of the released exosomes [96]. In Table 6, we demonstrate the complications of chronic hepatobiliary diseases.

In the next paragraphs, we will shed light on the implication of EVs in hepatobiliary malignancies and their potential role as biomarkers [97].

### 3.10. Gallbladder Cancer (GC)

Although a wide variety of studies exist regarding the development of novel biomarkers for several malignancies, there is a noticeable lack of studies specifically regarding GC [98]. It has been recently demonstrated by M. Kong et al. that there are aberrations in exosomes between GC patients and patients with other gallbladder pathologies (cholecystitis, gastric polyps) and healthy controls [99]. They demonstrated that exosomal membrane integrity was altered (lost) due to a noticeable decrease in several unsaturated phosphatidylcholines/phosphatidylethanolamines in GC patients, compared to other groups [100]. Meanwhile, Ueta, E. et al. demonstrated that circulating serum EV-miRNAs constitute novel GC biomarkers, as there are several alterations between EV-miRNAs in GC and non-GC patients [101]. Some of the EV-miRNAs that they demonstrated are miR-451a- and miR-1246-based in silico. The former was noticeably decreased, whereas the latter increased [101]. Additionally, the first one is implicated in the apoptosis pathway and GC suppression of proliferation via suppression of cyclin-dependent kinase inhibitor 2D (CDKN2D), MIF, and PSMB8, having a prognostic value, whereas the second is in tumor progression, proliferation, as well as invasion [99,100,101]. The second one, if it is combined with the neoplastic biomarkers carbohydrate antigen 19-9 (CA19-9) and carcinoembryonic antigen CEA, provides high diagnostic accuracy (AUC of 0.816), which can serve as a diagnostic GC biomarker [99,100,101]. In Table 7, we demonstrate some of the EV-based biomarkers for GC.

### 3.11. Hepatocellular Cancer

The emerging role of EVs in HCC pathogenesis is in the spotlight of scientific research, as it is the leading cause of cancer-related mortality globally. EVs have a key role in the intercellular communication, which eventually leads to hepatocarcinogenesis and tumor progression via altering the anti-cancer immune responses and cell survival [102,103]. EV-cargoes such as miRNAs and other non-coding RNAs (lncRNAs, circular RNAs (circRNAs), etc.) have a crucial role in HCC proliferation, progression, invasion, migration, and lastly metastatic dissemination via stimulating several signaling pathways such as PI3K/AKT and MAPK [104,105]. Moreover, EV-oncogenic protein and RNA molecules are significantly implicated in neoangiogenesis, promoting HCC typical hypervascularity, increased vascular permeability, and eventually migration and distant dissemination [104]. HCC-derived EV-miR-103 and miR-210 interact with endothelial cells, leading to their altered integrity and signal transducer and activator of transcription 6 (STAT6) and SMAD family member 4 (SMAD4) overexpression, respectively [106]. HCC-derived EVs interact with several other types of recipient cells in the tumor microenvironment (TME), altering their functional state through a paracrine and/or autocrine mode [104,105]. More particularly, HCC-derived EVs can promote EMT in TME via their embedded miR-3129, which has as a target the thioredoxin-interacting protein (TXNIP) [107]. At the same time, the interaction between HCC-EVs and other recipient cells leads to pro-inflammatory cytokine and metalloproteinase (MMP-2 and 9) overproduction that promotes HCC cell proliferation and migration [105]. Additionally, tumor macrophages in TME receive large-sized EVs let-7b, leading to overproduction of IL-6, while MMP-2 is also overproduced via the HCC-derived-EV-CD147 [102]. EMT and distant dissemination are also promoted via EVs-miR92a-3p [108]. Moreover, TME-fibroblasts are activated (cancer-associated fibroblasts (CAFs)) in highly malignant HCC, with an increased metastatic behavior via HCC-derived miR-1247-3p that alters the expression (decrease) of Beta-1,4-Galactosyltransferase 3 (B4GALT3) [109]. Interestingly, these EVs promote CAF-related pro-inflammatory cytokine production (IL-6 and 8), stimulating the formation of lung pre-metastatic niche fibroblasts, via the EMT phenomenon [109]. Meanwhile, small EVs-circ-PTGR1 modify the homeostasis of TME, facilitating HCC cell invasion in the surrounding tissues and migration to blood circulation [110]. HSCs are also receiving EVs such as EV-miR-21, a phenomenon that leads to their transformation into CAFs and the oversecretion of several growth factors, such as MMP-2 and 9, VEGF, as well as TGF-b [111]. Furthermore, the interaction between HCC-derived EVs-miR-23a and adipocytes and vice versa has been related to HCC progression, proliferation, and migration [112]. Meanwhile, adipocytes as EV-parental cells promote HCC growth by stimulating HCC deubiquitination and inhibiting miR-34a [105,112,113]. HSCs also receive HCC-derived EV-miR-21, which leads to their transformation into cancer-supporting cells [105,112,113]. The same is also reported in macrophages through HCC-derived-miR-23a-3p, while macrophages are polarized (M2-phenotype) via EVs-lncRNA TUC339, resulting in cytokine overproduction, tumor progression, and “defective” phagocytosis [105,112,114]. The aforementioned EVs are considered one of the most abundant, released by HCC. In addition, when macrophages receive HCC-derived EVs-miR-23a-3p, they increase PD-L1 expression, which impairs T-cell anti-cancer immune response, leading to HCC immune escape [115]. Some other tumor-promoting EVs’ non-coding RNAS are EV-miR-221, EV-miR-429, EV-CircFBLIM1, as well as EV-lncRNA FAL1 and EV-miR-25. The former has as a target p27/Kip1 in HCC, promoting its overexpression, the second targets Rb-binding protein 4 (RBBP4), promoting the expression of POU class 5 homeobox 1 (POU5F1), while the third is implicated in miR-338 and low-density lipoprotein receptor-related protein 6 (LRP6) axis, enhancing HCC progression [112,113,114,116]. The fourth is closely implicated in the gene expression of ZEB1 and alpha-fetoprotein (AFP), via targeting and competing miR-1236, whereas the fifth is implicated in HCC-resistance in sorafenib. Meanwhile, focusing on the EV-protein cargoes, HCC-derived-EV-CD147 and EV-complement factor H constitute tumor-promoting EVs, with the former interacting with fibroblast activation, while the latter with enhancing of inflammatory reaction and HCC progression via the overproduction of complement 5a and 3a [112,113,114,116].

Nevertheless, some of the tumor-suppressing EV cargoes are Huh7-derived EV-miR-122, which, when received by HepG2 cells in vitro, leads to the overproduction of IGF-1 and has a tumor-suppressive effect [112,113,114,117]. Likewise, EVs-EVs-circ-0051443 exhibit a HCC growth suppressive role via apoptosis promotion, as they act as sponges for miR-331-3p [112,113,114,117]. Additionally, HCC cells-derived EVs’ VEGF-suppressing proteins limit neoangiogenesis via AMPK signaling pathway activation, while EV-CLEC3B also suppresses EGF that is also implicated in angiogenesis [118]. Moreover, CAF-derived EV-miR-320a suppresses HCC proliferation and growth via targeting and inhibiting PBX3/ERK1/2/CDK2 signaling pathway in malignant hepatocytes [119]. Neoangiogenesis is also promoted by CD90-positive HCC-derived EV-H19 (lncRNA) via stimulating VEGF and VEGF receptor (VEGFR) overexpression in human umbilical vein endothelial cells (HUVECs) [120]. PI3K-Akt pathway, which has a key role in tumor progression and distant metastasis, is suppressed via HCC-derived EV-vacuolar protein sorting-associated protein 4A (Vps4A) [121]. In addition, it has been demonstrated that normal liver cells release SUMO-specific protease 3–eukaryotic initiation factor 4A1 complex (SENP3-EIF4A1) embedded in small-sized EVs. These act as tumor suppressors via decreasing the expression of miR-9-5p in the malignant hepatocytes [122]. In Table 8, we present a concise overview of EV-related HCC pathogenesis, while in Table 7, we demonstrate some of the EV-based biomarkers for HCC.

### 3.12. Cholangiocarcinoma (CCA)

CCA constitutes a highly aggressive malignancy, which arises in the biliary tract, encompassing a group of malignancies based on the anatomical site of the malignant lesion: intrahepatic CCA (iCCA), perihilar CCA (pCCA), and distal CCA (dCCA). Despite the current diagnostic and therapeutic advances, there are limited options regarding the therapeutic modalities, while CCA diagnosis is usually in already established advanced stages, leading to a poor prognosis [123]. EVs have a key role in the intercellular communication between CCA cells and the TME cells in their vicinity, leading to CCA cell proliferation, growth, and metastatic dissemination. As it is widely known, the implication of TME components is pivotal in CCA progression. Desmoplasia is a key characteristic of CCA, which includes the presence of a strongly fibrotic stroma around the tumor, where ECM modifications take place, as well as overproduction of pro-inflammatory molecules, cytokines, and tumor-promoting molecules (periostin, metalloproteinases, osteopontin, tenascin-C, etc.). CCA-derived EVs are implicated in stromal modification, as well as in the impaired anti-neoplastic immune responses and tumor escape phenomenon. Tumor escape is also attributed to CCA-derived EVs, which alter cytokine-induced killer cells (CIKs) expression in vitro, a phenomenon that alters their secretion of TNF-a and perforin [84]. Tumor migration and invasive behavior are facilitated by several CCA-derived EVs, such as EV-integrin-a or -b, EV-frizzled class receptor 10 (FZD10), EV-vitronectin, and EV-lactadherin, which alter several signaling pathways in the recipient cells. More particularly, the β-catenin pathway (Wnt signaling pathway), which regulates cell proliferation, and differentiation and can potentially lead to carcinogenesis, is significantly modified by vitronectin, lactadherin, and integrin-a/b that are embedded in CCA-derived EVs, leading to the overexpression of β-catenin. This phenomenon leads not only to increased CCA migration and invasion but also to proliferation and distant metastasis [124]. Moreover, it has been demonstrated that iCCA-derived EV-ceramide and/or dihydroceramide are highly found in CCA patients, while they are implicated in blood dissemination of tumor cells in cases of the latter and monocyte cytokine overproduction in the case of the former [125]. Additionally, both types of EVs are derived from iCCA, with poor differentiation EVs that contain circ-CCAC1 are highly detected in bile and malignant tissue specimens, being deeply involved in neoangiogenesis as they interact with endothelial cells, as well as in Yin Yang 1 (YY1) overregulation via binding and sponging of miR-514a-5p. The overexpression of YY1 has been closely related to the migratory behavior of CCA cells, as well as to distant metastasis [83,84,126]. In addition, CCA-derived-EV-miR-183-5p induces overexpression of VEGF by mast cells, promoting neoangiogenesis, as well as protumorigenic prostaglandin E2 (PGE2) and prostaglandin E receptor 1 (PTGER1) [81,82,97,124,125].

Similarly, CCA-derived Circ-0000284 sponges miR-637, leading to LY6E upregulation, which eventually results in the malignant transformation of physiological cholangiocytes [126,127]. It is observed that HuCCT1-derived EVs, which are a population of vesicles that are derived from the HuCCT1 CCA line, highly contain CXCL-1, alpha-smooth muscle actin (α-SMA), vimentin, fibroblast activation protein, CCL2, as well as IL-6, with the latter acting as a growth factor for CCA. Meanwhile, the intercellular interactions between HUCCT1-derived EVs and MSCs in vitro lead to CCA progression [128]. MSCs are transformed into CAFs after interacting with CCA-derived EVs, leading to desmoplasia. Moreover, TAM-derived EVs-circ-0020256 enhance CCA proliferation, migration, and dissemination. T-CD8-positive cells’ chemotaxis is also modulated in CCA, via EV-B-cell-specific Moloney murine leukemia virus integration site 1 (BMI1), which is significantly elevated in the malignant tissue, leading to CCA growth, progression, migration, and metastasis, constituting a possible druggable target [81,128,129]. Furthermore, it has been demonstrated that there are several other CCA-derived-EV-miRNAs that are highly expressed in CCA, such as EV-miR-34c, miR-183-5p, as well as miR-200c-3p and miR-200b-3p [124]. The last two are significantly higher in CCA, proportionally to the tumor stage; the second is implicated in iCCA chemoresistance, as it induces PD-L1 overexpression on the surface of macrophages, while the former is related to increased CCA progression and growth. On the other hand, there are some tumor-suppressive vesicles, such as EV-miR-30e, which suppress EMT in CCA, impeding CCA cell dissemination and their invasive behavior. HSC-derived EV-miR-195 also acts as a tumor suppressor, inhibiting CCA progression, while EV-miR-195 also suppresses CCA cell growth in vitro [130]. In Table 9, we present EV-related CCA pathogenesis, while in Table 7, we demonstrate some of the EV-based biomarkers for CCA.

## 4. Limitations of EVs Utilization as Biomarkers

Exosomes, a subtype of EVs, are promising noninvasive biomarkers for chronic hepatobiliary diseases [82,97]. These vesicles are released into various body fluids—such as blood, bile, and urine—and carry biomolecules like proteins, lipids, and nucleic acids, reflecting the pathological state of the cells or tissues from which they originate. A major advantage of EVs lies in their potential for early diagnosis [114]. Their cargo often undergoes molecular changes before symptoms appear or conventional biomarkers are altered. Their dynamic content makes them useful for monitoring disease progression and treatment response. Furthermore, their lipid bilayer offers protection from enzymatic degradation, which enhances their stability in circulation [131]. Despite their promise, several limitations remain. However, a major challenge is the lack of standardized methods for EV isolation and characterization (Table 10). Techniques like ultracentrifugation, immunoaffinity capture, filtration, and size-exclusion chromatography produce variable results in purity and subtype, affecting study reproducibility [114,132]. EVs are also heterogeneous in size, origin, and molecular cargo, which complicates functional analysis and data interpretation. Additionally, overlapping molecular signatures between different diseases can reduce biomarker specificity [81,114,132,133]. Analytical tools used to analyze EVs are often complex and not widely available in clinical settings. Storage conditions are also critical—freezing at −80 °C is recommended to maintain EV structure and molecular content, as repeated freeze–thaw cycles degrade particles and RNA. Stabilizers like trehalose may help reduce damage [131]. From a regulatory standpoint, there is no unified framework for classifying EV-based products, whether as diagnostics, therapeutics, or biologics. This ambiguity complicates clinical approvals. Ethical concerns also arise, particularly with engineered or donor-derived EVs, regarding donor safety, consent, and traceability [134]. Contaminants like protein aggregates and lipoproteins can co-purify with EVs, potentially skewing results from omics-based analyses [135]. Another major limitation is the incomplete understanding of EV bio-distribution, uptake, and clearance in vivo—critical factors for designing therapeutic delivery systems. On the production side, large-scale manufacturing of EVs faces issues like low yield, batch variability, and lack of GMP standards [136]. Moreover, current isolation methods are related to resources, implying several limitations such as the cost and infrastructure, with the standardization of these being necessary for clinical laboratories. Integration of EV analysis in combination with existing diagnostic modalities, such as biopsy, serological lab tests, and imaging modalities, that will provide a multi-modal interpretation, is considered a challenge as it requires an algorithmic approach and harmonization. In addition, the common EV profiling in several diseases constitutes another challenge for the interpretation of EV findings in the presence of multiple diseases in each patient (e.g., chronic hepatitis B and MASLD), so the development of disease-specific panels may overcome this issue. Finally, inconsistencies in EV definitions, characterization, and reporting across studies hinder reproducibility and consensus. In conclusion, while EVs show great potential as early, dynamic, and tissue-specific biomarkers for chronic liver diseases, several biological, technical, and regulatory challenges must be resolved before they can be widely adopted in clinical practice. Standardization, mechanistic understanding, ethical regulation, and clinical validation are all key steps needed for successful translation [114,131,132,133,134,135].

## 5. Conclusions and Future Directions

EVs can potentially serve as non-invasive biomarkers in hepatobiliary disease, especially due to their wide variety of cargoes that can be potentially targeted for multi-parametric analysis. The standardization of EV isolation, characterization, and storage protocols may increase the performance of EV-based diagnostics, while the combination of multi-omics findings may open new horizons in the development of novel biomarkers and may shed light on the complexity of hepatic and biliary pathologies. Finally, in addition to EVs’ role as diagnostic and prognostic biomarkers, they have an emerging role as promising tools for targeted therapies, due to their ability to transfer several cargoes. This ability opens new horizons for future perspectives in EV research and engineering for drug delivery and personalized hepatobiliary disease management.

## Figures and Tables

**Figure 1 ijms-26-06333-f001:**
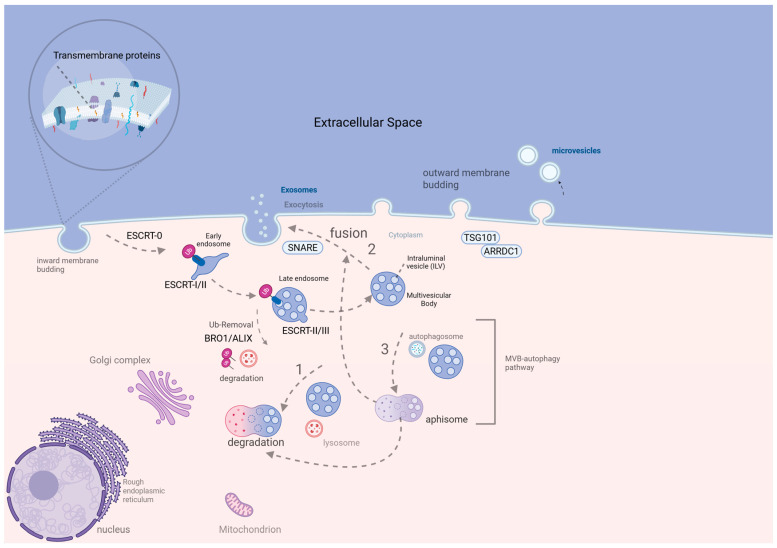
Microvesicle and ESCRT-dependent exosome biogenesis [7]. The exosome biogenesis starts with the internalization of transmembrane proteins under a close dependence on ESCRT 0-III for cargo recruitment. ESCRT-0 (Vps27, Hse1) identifies the internalized ubiquitinated transmembrane proteins. The binding of Vps27 on the endosomal membrane permits the binding of ESCRT machinery on the endosome. ESCRT-0 binds to ESCRT-I (Vps23, Vps28) and the latter on ESCRT-II (Vps25, Vps36). Then, ESCRT-II binds and recruits ESCRT-III (Vps20), followed by the formation of ILVs (budding of late endosomal membrane), which are integrated into the lumen of MVB. On that point, MVB final formation requires the removal of Ubiquitin (Up) tags by Bro1/ALIX proteins (part of the ESCRT complex). Once MVB is formed, it can either be (1) fused with lysosome for degradation, (2) fused with plasma membrane for exosome exocytosis under SNARE proteins contribution, or (3) fused with autophagosome, forming amphisome, which is either degraded or fused with the cell membrane for exosome release. Microvesicles result from the outward blebbing of the plasma membrane under the TSG101-ARRDC1 interaction [7], created in BioRender. Trifylli, E. (2025) https://BioRender.com/e99v625 (accessed on 4 May 2025) (Agreement number EM28FVFH0Z). ALIX, ALG-2-interacting protein X; ARRDC1, arrestin domain-containing protein 1; BR01/BRO1, Bro1 domain protein; ESCRT, endosomal sorting complex required for transport; EV, extracellular vesicle; ILV, intraluminal vesicle; MVB, multivesicular body; SNARE, soluble NSF attachment protein receptor; TSG101, tumor susceptibility gene 101; Ub, ubiquitin; Vps, vacuolar protein sorting protein.

**Table 1 ijms-26-06333-t001:** EV-based diagnostic biomarkers for chronic hepatitis B and C.

Biomarker	Description
Serum EV-KV311 [21]	Increased levels in chronic hepatitis B versus HCC patients
Serum EV-CO9/SVEP1 [21]	Increased levels in chronic hepatitis B compared to healthy controls
Serum EV-LBP [21]	Decreased levels in chronic hepatitis B versus HCC patients
Serum EV-Willebrand factor [21]	Decreased levels in chronic hepatitis B versus cirrhotic patients
Plasma EV-hsa-miR-221/hsa-miR-1290 [22]	Downregulated in HCV-HIV coinfection

CO9, complement component 9; LBP, lipopolysaccharide binding protein; HCC, hepatocellular carcinoma; HCV, hepatitis C virus; HIV, human immunodeficiency virus.

**Table 2 ijms-26-06333-t002:** (**a**). Effects of EV-cargoes in MASLD pathogenesis. (**b**). EV-based biomarkers in MASLD/MASH.

(a)			
EV Cargo	Parental Cell	Target Cell(s)	Effect(s)
Vanin-1 [19]	Lipotoxic hepatocytes	LSECs, HSCs	Increases angiogenesis (LSECs) and fibrogenesis (HSCs)
miR-1 [39]	Lipotoxic hepatocytes	LSECs	Promotes angiogenesis and disease progression
let-7e-5p [40]	Lipotoxic hepatocytes	Pre-adipocytes (adipose tissue)	Alters lipid deposition, promotes lipogenesis
CXCL10 [41]	Lipotoxic hepatocytes	Monocytes/Macrophages	Enhances chemotaxis, inflammation (inhibited via MLK3)
TRAIL [42]	Lipotoxic hepatocytes	Monocytes/Macrophages	Activates macrophages, NF-κB-mediated immune response
Mitochondrial DNA (oxidized) [43]	Lipotoxic hepatocytes	Monocytes/Macrophages	Enhances chemotaxis, macrophage recruitment
Ceramides [26,44]	Lipotoxic hepatocytes	Monocytes/Macrophages	Enhances chemotaxis, macrophage recruitment
miR-192-5p [45]	Lipotoxic hepatocytes	Monocytes/Macrophages	Increases chemotaxis
Integrin-β1 [46]	Lipotoxic hepatocytes	Monocytes/Macrophages	Enhances inflammatory response; elevated in MASH with fibrosis
miR-128-3p miR-192 [47]	Lipotoxic hepatocytes	HSCs	Activates HSCs, enhances fibrogenesis
Sphingosine kinase 1/S1P [48]	Monocytes, HSCs, LSECs	HSCs	Activates HSCs, enhances fibrogenesis
VEGF [49]	Portal fibroblasts	Endothelial cells	Enhances angiogenesis
Pro-fibrogenic cytokines: TGF-β, CTGF, PDGF [50]	Kupffer, LSECs	HSCs	Enhance fibrosis and angiogenesis, ECM production
miR-128-3p [51]	Lipotoxic hepatocytes	HSCs	PPAR-γ inhibition in HSCs, activation of HSCs Increased pro-fibrogenic genes expression (a-SMA, TIMP-2, collagen type I)
miR-1297 [52]	Lipotoxic hepatocytes	HSCs	Effect on PTEN/PI3K/AKT signaling in HSCs, HSCs activation, and enhanced fibrogenesis
Hedgehog ligands [53]	Activated HSCs	LSECs	LSECs capillarization Vascular remodeling Enhanced fibrogenesis
LIMA1 protein [54]	Lipotoxic hepatocytes	HSCs	Mitophagy suppression HSC activation increased COL1A1/A3, α-SMA
LSEC-derived EVs [55]	Healthy LSECs	HSCs Kupffer cells	Decrease in HSC/Kupffer cells activation. For instance, EVs from healthy cells
**(b)**			
**Biomarker**	**Description/Relevance**		**Diagnostic Accuracy (AUC, Sensitivity (%), Specificity (%))**
ITGβ1 [46]	From LPC-treated hepatocytes; promotes macrophage infiltration and inflammation in MASH. Suppressed by antibodies		-
EV-S1P [36,48]	From hepatocytes; involved in fibrogenesis via HSC activation, increased in fibrosis		-
EV-TRAIL [42]	Implicated in MASH; suppression limits disease progression in animal models		-
EV-ASGPR1 [56]	Increased MASLD with advanced fibrosis		0.83
EV-SLC27A5 [57]	Elevated in advanced MASLD and MASLD-HCC		-
EV-miR-22 [58]	Levels increase proportionally with MASLD severity		
EV-miR-16 [29] mir-128-3p [47,51,58], mir-192-5p [58], mir-129 [29,58]	Elevated in MASLD/MASH; not liver-specific		-
miR-574-3p, miR-542-3p, miR-200a-3p [59]	Elevated in MASLD patients		-
miR-542-3p, and miR-200a-3p [59]	Elevated in MASLD patients with advanced fibrosis		-
EV-miR-122 [60]	Hepatocyte-specific (70% expression); elevated in advanced MASLD and extended hepatic injury		0.77
EV-miR-34a [60]	Non-liver-specific; elevated in MASLD/MASH with fibrotic injury		-
EV-heat shock proteins [61]	Reflect stress-induced hepatocyte injury		-
Leukocyte-derived EVs [62]	Inversely correlated with hepatic fibrosis severity		-
miR-135a-3p miR-122-5p miR-504-3p	Significantly reduced in the serum of MASLD patients vs. healthy controls		miR-135a-3p: 0.849 vs. ALT (0.672) miR-122-5p: 0.790 miR-504-3p: 0.708
Total circulating EVs (post-operation) [63]	Significantly reduced after bariatric surgery and weight loss		-
miR-21-5p, miR-151a-3p, miR-126-5p	Liver stiffness, steatosis evaluation		0.76–0.81 0.95 (miR-126-5p + leptin): best for steatosis 0.81 (miR-151a-3p + glucose)

(a): CTGF, connective tissue growth factor; CXCL10, C-X-C motif chemokine ligand 10; ECM, extracellular matrix; HSCs, hepatic stellate cells; ITGβ1, integrin beta 1; LSECs, liver sinusoidal endothelial cells; MLK3, mixed-lineage kinase 3; PDGF, platelet-derived growth factor; TGF-β, transforming growth factor beta; TRAIL, TNF-related apoptosis-inducing ligand; VEGF, vascular endothelial growth factor. (b): ASGPR1, asialoglycoprotein receptor 1; ITGβ1, integrin beta 1; MASLD, metabolic dysfunction-associated steatotic liver disease; MASH, metabolic dysfunction-associated steatohepatitis; S1P, sphingosine-1-phosphate; SLC27A5, solute carrier family 27 member 5; TRAIL, TNF-related apoptosis-inducing ligand.

**Table 3 ijms-26-06333-t003:** EV-based biomarkers in ALD and AH.

Biomarker	Description/Relevance	Diagnostic Accuracy (AUC; Sensitivity (%); Specificity (%))
EV-CYP2E1 [67]	Diagnostic marker of hepatocyte injury in chronic alcohol exposure; linked to ER and oxidative stress, and monocyte toxicity	
Plasma EV-miR-19b [66,67]	Elevated in alcohol-associated liver fibrogenesis models	
Plasma EV-sphingolipids [72]	Levels correlate with AH severity; serve as prognostic biomarkers	
Plasma EV-CK18 (M30, M65) [73]	Diagnostic markers for AH	EV-CK18 M30: 0.75–0.863; 71.5%; 84.6% EV-CK18 M65: 0.82–0.91; 75%; 76% better for inflammation
EV-ASGPR1 and EV-CD34+ [56]	Increased corticosteroid non-responders in AH; function as predictive biomarkers for treatment response	

AH, alcoholic hepatitis; ASGPR1, asialoglycoprotein receptor 1; AUC, area under the curve; CK18, cytokeratin 18; CYP2E1, cytochrome P450 2E1.

**Table 4 ijms-26-06333-t004:** EVs in iron homeostasis and HH diagnosis [76,77,78,79].

Topic	Key Component/Process	Function/Mechanism	Outcome/Relevance
**EVs in Iron Regulation**	Ferritin-containing EVs	EVs carry ferritin in circulation and urine	Non-invasive biomarker of iron status
	Mitochondria-derived EVs	May deliver ferritin to recipient cells or back to circulation via the multivesicular body–exosome pathway	Potential iron redistribution mechanism
	Bone marrow-derived EVs	Modulate hepcidin production	Involved in systemic iron homeostasis
	EVs under oxidative stress	Carry antioxidant proteins related to iron metabolism	Mitigate ROS damage; modulate ferroptosis
	EV-mediated iron redistribution	EVs sequester excess iron from parental cells	Protect parental cells and may harm recipient cells
**EVs in Hepatic Iron Overload**	EVs from hepatocytes/macrophages	Altered ferritin and iron-handling enzyme expression	Reflect intracellular iron load; signal local and systemic stress
	Macrophage-derived EVs	High ferritin content	Indicator of iron overload and inflammation
	Hepatocyte-derived EVs in blood/bile	Reflect liver iron load and damage	Useful in MASLD, HH, and other hepatic conditions
	Hepatocyte-derived EVs as biomarkers	Reflect liver stress and iron imbalance	Track disease progression in HH and related disorders
	EV-iron handling enzymes	Reflect intracellular iron status	EVs can serve as sensitive markers of iron-rich conditions, such as HH
	EV-ferritin released by hepatocytes/macrophages	Reflect intracellular iron status	Diagnostic markers of intracellular iron status, non-invasive monitoring biomarkers for HH-related liver injury

HH, hereditary hemochromatosis; MASLD, metabolic dysfunction-associated steatotic liver disease; ROS, reactive oxygen species.

**Table 5 ijms-26-06333-t005:** The role of EVs as biomarkers for autoimmune hepatobiliary diseases.

Condition	EV Component/Origin	Target/Function	Outcome/Associated Effect
PSC [84]	Serum EV-lncRNA H19	Correlates with PSC severity	Fibrogenesis; disease progression
PBC and PSC [82]	Cholangiocyte-derived EVs	Involved in bile duct homeostasis and intercellular crosstalk	Dysregulated in cholangiopathies
	Hepatocyte-EVs (EGFR and ITGB4)	Biliary tract oncogenesis	Oncogenic signaling
	Bile EVs to cholangiocyte cilia	Induce miR-15a → suppress ERK signaling	Inhibit cholangiocyte proliferation
AH [85,86]	MSC-exosomes-miR-21 and miR-16	Promote pro-inflammatory macrophage phenotype	Can worsen the inflammatory profile
	MSC-exosomes	Inhibit T-cell proliferation and migration (↓ CCL1, CCL2, CCL21)	Reduced chemotaxis, immune suppression
	MSC-EVs	Promote Th1 and Th2 transition	Anti-inflammatory shift
	MSC-EVs	PD-L1 expression T-cell function inhibition	Immune suppression

AH, alcoholic hepatitis; CCL1, C-C motif chemokine ligand 1; CCL2, C-C motif chemokine ligand 2; CCL21, C-C motif chemokine ligand 21; EGFR, epidermal growth factor receptor; ERK, extracellular signal-regulated kinase; ITGB4, integrin beta 4; lncRNA, long non-coding RNA; MSC, mesenchymal stem cell; PBC, primary biliary cholangitis; PD-L1, programmed death-ligand 1; PSC, primary sclerosing cholangitis; Th1, T helper cell type 1; Th2, T helper cell type 2.

**Table 6 ijms-26-06333-t006:** The complications of chronic hepatobiliary diseases [26,87,88,89,90,91,92,93,94,95,96,97].

Complication	Description	EV-Related Mechanisms
Fibrotic injury and cirrhosis [50,87,89]	Excessive ECM deposition leading to architectural liver distortion and angiogenesis.	↑ EV-PDGFRα in circulation; ↓ EV-Twist1 and miR-214 from HSCs; hepatocyte-derived EV-miR-128-3p and miR-192 activate HSCs via PPARγ suppression; LSEC-EV-SK1 promotes fibrosis.
PH, hepatopulmonary and porto-pulmonary syndromes [7,90,91,92]	Results from vascular resistance and fibrosis; causes systemic vasodilation and pulmonary complications.	Large EVs involved in vasodilation (↑ in CP B/C); small EVs activate JAK2/ROCK → ↑ resistance; ↑ EV-miR-194 in hepatopulmonary syndrome; EV-VEGF from portal myofibroblasts worsens the condition.
Coagulation disorders [19,26,49,93]	Altered coagulation due to liver dysfunction, leading to bleeding or thrombosis.	Platelet-derived EV-annexin V ↑ in severe cirrhosis; EV-tissue factor promotes clotting; HSC, cholangiocyte, and hepatocyte EVs enhance angiogenesis; LSEC-derived EV-VEGF also promotes angiogenesis.
Ascites and HE [19,94]	Fluid accumulation and cognitive impairment in advanced liver disease.	↑ Hepatocyte and endothelial EVs in ascites → associated with mortality; altered EV protein cargoes in HE models; small vesicles in ascites promote inflammation.
Biliary tract stenosis and chronic cholecystitis [95,96]	Narrowing of bile ducts and gallbladder inflammation; can progress to malignancy.	Bile-EV levels (e.g., Severino et al.) distinguish malignant vs. benign CBD stenosis (100% accuracy); EVs from microbial-infected cells carry bacteria → dysbiosis and cholecystitis; exosomes modulate gene expression and inflammatory signaling.
Hepatobiliary malignancies [97]	Includes HCC, CCA, GBC; often the final stage of chronic hepatobiliary disease.	Multiple EV biomarkers involved (e.g., miRs, lncRNAs, circRNAs, proteins, and lipids); EVs mediate tumor progression, immune evasion, angiogenesis, and chemoresistance.

CBD, common bile duct; CP, Child–Pugh; circRNAs, circular RNAs; ECM, extracellular matrix; HE, hepatic encephalopathy; HSCs, hepatic stellate cells; lncRNAs, long non-coding RNAs; LSEC, liver sinusoidal endothelial cell; PDGFRα, platelet-derived growth factor receptor alpha; PPARγ, peroxisome proliferator-activated receptor gamma; PH, portal hypertension; SK1, sphingosine kinase 1; Twist1, Twist family BHLH transcription factor 1; VEGF, vascular endothelial growth factor.

**Table 7 ijms-26-06333-t007:** EV-based biomarkers for GC, CCA, and HCC.

Cancer Type	Biomarker	Description/Relevance	Diagnostic Power AUC; Sensitivity(%); Specificity (%)
GC [82,98,99,100,101]	EV-miR-451a	Decrease in GC; involved in apoptosis and tumor suppression via CDKN2D, MIF, PSMB8. Prognostic marker.	0.664; 62.0%; 75.0%
	EV-miR-1246	Increased GBC; promotes tumor progression, invasion, and proliferation.	0.646; 60.0%; 66.7%
		CEA + CA19-9 + miR-1246 in serum EVs	0.816; 72.0%; 90.8%
		vs. CEA	0.770; 60.0%; 83.3%
		CA19-9	0.729; 58.0%; 92.6%
HCC [102,103,104,105,106,107,108,109,110,111,112,113,114,115,116,117,118,119,120,121,122]	EV-miR-224, -221, -21, -665, -222, -18	Elevated in HCC; diagnostic markers.	EV-miR-224: 92.5%; 90%; accuracy 94% EV-miR-221: 0.880; 86.5%; 76.7% EV-miR-21: 0.773; 61.1%; 83.3% EV-miR-222 and EV-miR-221: 0.84; 86%; 66% vs. cirrhosis/chronic hepatitis
	EV-miR-101, -125b	Decreased; miR-125b associated with poor survival and recurrence.	EV-miR-101 0.956; 92.5%; 97.5% EV-miR-125b: 0.739; 83%; 67.9% for recurrence, 0.702; 82.5%; 53.4% survival
	EV-hsa_circ_0028861	Distinguishes HCC vs. CHB	0.83; 76.79 %; 78.95 %
	EV-miR-21-5p + AFP	Elevated in plasma; diagnostic and monitoring value.	0.85;95%;50% vs. AFP
	EV-miR-93	Poor prognosis; promotes progression via TIMP2, CDKN1A, TP53INP1 targeting.	
	EV-LINC00853	Distinguish AFP (−) early HCC	0.883
		Distinguish AFP (+) early HCC.	0.897
	EV-miR-25	Associated with resistance to sorafenib therapy.	
	EVs-SH3BGRL3, ANGPTL3, IFITM3	Elevated in viral-HCC; potential early diagnostic biomarkers.	
	EV-miR-92b	Elevated post-transplant in recurrent HCC. Predictive biomarker.	
	EV-miR-148a	Differentiate HCC vs. cirrhosis.	0.891
	EV-miR-19–3p	Distinguishing non-hepatitis B, non-hepatitis C-infected HCC.	0.82
	EV-miR-19-3P ± AFP		0.92
	EV-lnc85	Differentiate HCC vs. cirrhosis.	0.888
	EV-DANCR	Predicting post-operative recurrence in HCV-infected HCC patients.	0.88
CCA [81,82,122,123,124,125,126,127,128,129,130]	EV-FZD10	Promotes CCA growth/metastasis; a recurrence predictor.	
	EV-ceramide/dihydroceramide	Correlated with tumor progression and poor prognosis.	
	Exosomal membrane lipids (PCs/PEs)	Decreased unsaturated phosphatidylcholines and phosphatidylethanolamines in GC, CCA; associated with loss of membrane integrity. Potential diagnostic marker.	0.857; 71.4%; 100% (CCA vs. benign) and assay kit phosphatidylcholine: 0.839; 71.4%; 100%

AFP, alpha-fetoprotein; AUC, area under the curve; CCA, cholangiocarcinoma; CA19-9, carbohydrate antigen 19-9; CEA, carcinoembryonic antigen; CHB, chronic hepatitis B; EV, extracellular vesicle; GBC, gallbladder cancer; HCC, hepatocellular carcinoma; HCV, hepatitis C virus; IFITM3, interferon-induced transmembrane protein 3; lnc85, long non-coding RNA 85; LINC00853, long intergenic non-protein coding RNA 853; miR, microRNA; PC, phosphatidylcholine; PE, phosphatidylethanolamine; SH3BGRL3, SH3 domain-binding glutamic acid-rich-like protein 3; TIMP2, tissue inhibitor of metalloproteinases 2; TP53INP1, tumor protein p53 inducible nuclear protein 1; VEGF, vascular endothelial growth factor; YY1, yin yang 1 transcription factor.

**Table 8 ijms-26-06333-t008:** A concise overview of EV-related HCC pathogenesis.

Specific EV Components	Mechanism/Pathway	HCC Pathogenesis Role	Mechanism
miR-103, miR-210 [106]	STAT6, SMAD4 overexpression	Altered endothelial integrity	HCC-EV-miR-103, miR-210 affect endothelial cells.
miR-3129 [107]	EMT induction	Promotes metastasis	EV-miR-3129 targets TXNIP.
MMP-2, MMP-9 [105]	Pro-inflammatory signals	Proliferation, migration	EVs trigger cytokine and MMPs release.
let-7b, CD147 [102]	IL-6, MMP-2 upregulation	Inflammation, invasion	Macrophages uptake EV-let-7b and CD147.
miR-92a-3p [108]	EMT pathway	Metastasis	EV-miR-92a-3p promotes EMT.
miR-1247-3p [109]	Downregulates B4GALT3	CAF activation, metastasis	EV-miR-1247-3p affects CAFs via B4GALT3.
CAF-EVs [109]	Inflammatory cytokines	Lung pre-metastasis niche	CAFs produce IL-6, IL-8, inducing lung niche.
circ-PTGR1 [110]	TME homeostasis disruption	Invasion, migration	EV-circ-PTGR1 modifies TME.
miR-21 [105,111,112,113]	Growth factor overproduction	CAF transformation	HSCs uptake EV-miR-21.
miR-23a [105,112,113]	Adipocyte crosstalk	Proliferation, migration	EV-miR-23a interacts with adipocytes.
Adipocyte-EVs [105,112,114]	Deubiquitination	HCC growth promotion	Adipocyte EVs suppress miR-34a.
miR-23a-3p, TUC339 [105,112,114]	M2 polarization, PD-L1 ↑	Immune escape, cytokine release	Macrophages uptake EV-miR-23a-3p, TUC339.
miR-221 [112,113,114,117]	Cell cycle regulation	Proliferation	EV-miR-221 targets p27/Kip1.
miR-429 [112,113,114,117]	Gene expression control	Stemness, progression	EV-miR-429 affects RBBP4 and POU5F1.
circFBLIM1 [112,113,114,117]	Wnt/β-catenin axis	Proliferation	EV-circFBLIM1 involves miR-338/LRP6.
FAL1 [112,113,114,117]	ZEB1, AFP expression	Metastasis, tumor marker ↑	EV-lncRNA FAL1 targets miR-1236.
miR-25 [112,113,114,117]	Drug resistance	Therapy resistance	EV-miR-25 mediates sorafenib resistance.
CD147, CFH [112,113,114,117]	Fibroblast, inflammation	Tumor progression	EV-CD147 and complement factor H are oncogenic.
miR-122 [112,113,114,117]	IGF-1 modulation	Tumor suppression	EV-miR-122 from Huh7 suppresses the tumor.
circ-0051443 [112,113,114,117]	Apoptosis promotion	Growth inhibition	EV-circ-0051443 sponges miR-331-3p.
VEGF-suppressors [118]	AMPK pathway	Angiogenesis suppression	EV-VEGF-suppressing proteins inhibit angiogenesis.
CLEC3B [118]	EGF signaling	Angiogenesis control	EV-CLEC3B suppresses EGF.
miR-320a [119]	Inhibits PBX3/ERK/CDK2	Growth suppression	CAF-derived EV-miR-320a targets PBX3/ERK1/2.
H19 [120]	VEGF/VEGFR ↑	Neovascularization	EV-H19 lncRNA promotes angiogenesis.
Vps4A [121]	Tumor signaling suppression	Metastasis inhibition	EV-Vps4A suppresses the PI3K-AKT pathway.
SENP3-EIF4A [122]	miR-9-5p suppression	Tumor suppression	Normal liver EVs carry SENP3-EIF4A1.

AFP, alpha-fetoprotein; AMPK, AMP-activated protein kinase; B4GALT3, beta-1,4-galactosyltransferase 3; CAFs, cancer-associated fibroblasts; CD147, cluster of differentiation 147; CDK2, cyclin-dependent kinase 2; CDKN2D, cyclin-dependent kinase inhibitor 2D; CLEC3B, C-type lectin domain family 3 member B; circRNAs, circular RNAs; EGF, epidermal growth factor; EMT, epithelial-to-mesenchymal transition; ERK1/2, extracellular signal-regulated kinase 1/2; EV, extracellular vesicle; HCC, hepatocellular carcinoma; HSCs, hepatic stellate cells; HUVECs, human umbilical vein endothelial cells; IGF-1, insulin-like growth factor 1; IL-6, interleukin 6; IL-8, interleukin 8; lncRNAs, long non-coding RNAs; LRP6, low-density lipoprotein receptor-related protein 6; MAPK, mitogen-activated protein kinase; MMP-2, matrix metalloproteinase-2; MMP-9, matrix metalloproteinase-9; miR, microRNA; PBX3, pre-B-cell leukemia homeobox 3; PD-L1, programmed death-ligand 1; PI3K/AKT, phosphoinositide 3-kinase/protein kinase B pathway; POU5F1, POU class 5 homeobox 1; Rb, retinoblastoma protein; RBBP4, Rb-binding protein 4; SENP3-EIF4A1, SUMO-specific protease 3–eukaryotic initiation factor 4A1 complex; SMAD4, SMAD family member 4; STAT6, signal transducer and activator of transcription 6; TGF-β, transforming growth factor beta; TXNIP, thioredoxin-interacting protein; TNF-α, tumor necrosis factor alpha; TME, tumor microenvironment; VEGF, vascular endothelial growth factor; VEGFR, vascular endothelial growth factor receptor; Vps4A, vacuolar protein sorting-associated protein 4A; ZEB1, zinc finger E-box-binding homeobox 1.

**Table 9 ijms-26-06333-t009:** EV-related CCA pathogenesis.

EV Role/Content	Key Concept	Mechanism/Effect
CCA-derived EVs [81]	Desmoplasia	Promotes fibrotic stroma, ECM remodeling, cytokine, and tumor-promoting molecule production
	Immune escape	Impairs anti-tumor immune responses, inhibits CIK function (TNF-α, perforin)
EV-integrin α/β, FZD10, vitronectin, lactadherin [124]	Migration/invasion	β-catenin/Wnt pathway activation, enhanced migration, proliferation, metastasis
EV-ceramide, dihydroceramide [125]	Distant dissemination	Blood dissemination, monocyte cytokine overproduction
EV-circ-CCAC1 [81,82,124]	Neoangiogenesis	Endothelial interaction, YY1 upregulation via miR-514a-5p sponging
EV-miR-183-5p [81,82,97,124,125]	Neoangiogenesis	Induces VEGF, PGE2, PTGER1 via mast cells
EV-circ-0000284 [128]	Oncogenic transformation	Sponges miR-637, upregulates LY6E
HuCCT1-EVs [128]	Tumor microenvironment	Contains CXCL-1, α-SMA, vimentin, FAP, CCL2, IL-6; induces CAFs from MSCs
TAM-derived EV-circ-0020256 [129]	Macrophage EVs	Enhances CCA proliferation, migration, and dissemination
EV-BMI1	Immune modulation	Modulates CD8+ T-cell chemotaxis, promotes progression
EVmiR-34c, miR-183-5p, miR-200c-3p, miR-200b-3p [124]	Oncogenic EV-miRNAs	Promotes growth, PD-L1 induction, and chemoresistance
EV-miR-30e	Tumor suppression	Suppresses EMT, inhibits dissemination
HSC-derived EV-miR-195 [130]	Tumor suppression	Inhibits growth and progression in vitro

α-SMA, alpha-smooth muscle actin; BMI1, B lymphoma Mo-MLV insertion region 1 homolog; CCA, cholangiocarcinoma; CAF, cancer-associated fibroblast; CCL2, C-C motif chemokine ligand 2; CIK, cytokine-induced killer; CXCL-1, C-X-C motif chemokine ligand 1; ECM, extracellular matrix; EMT, epithelial-to-mesenchymal transition; EV, extracellular vesicle; FAP, fibroblast activation protein; FZD10, frizzled class receptor 10; HSC, hepatic stellate cell; IL-6, interleukin-6; LY6E, lymphocyte antigen 6 family member E; MSC, mesenchymal stem cell; PD-L1, programmed death-ligand 1; PGE2, prostaglandin E2; PTGER1, prostaglandin E receptor 1; TAM, tumor-associated macrophage; VEGF, vascular endothelial growth factor; YY1, yin yang 1 transcription factor.

**Table 10 ijms-26-06333-t010:** Limitations in EV research [114,131,132,133,134,135].

Limitations	Description/Impact
**Isolation**	Lack of standard methods (e.g., ultracentrifugation, immunoaffinity, filtration, size-exclusion); affects purity and reproducibility.
**Heterogeneity**	EVs vary in size, origin, and cargo, complicating function and interpretation.
**Specificity**	Overlapping molecular signatures across diseases can reduce diagnostic specificity.
**Analytical complexity**	Advanced tools needed for analysis are often not available in clinical labs.
**Storage requirements**	Must freeze at −80 °C; avoid repeated freeze–thaw cycles; trehalose may help preserve integrity.
**Regulatory issues**	No unified framework for diagnostic/therapeutic classification; this complicates clinical approval.
**Ethical considerations**	Concerns about donor safety, consent, and traceability—especially for engineered/donor-derived EVs.
**Contaminants**	Risk of co-purifying protein aggregates and lipoproteins, skewing omics analyses.
**Knowledge gaps**	Incomplete understanding of in vivo biodistribution, uptake, and clearance.
**Manufacturing issues, costs, and resources**	Low yield, batch variability, absence of GMP standards hinder scalability, resource-dependent, high costs for manufacturing.
**Definition inconsistency**	Inconsistent EV definitions and reporting methods reduce reproducibility and scientific consensus.
**Lack of compatibility with other diagnostic modalities**	Technological harmonization is required, an algorithmic approach and multi-modal interpretation are needed.
**Needs for clinical translation**	Standardization, mechanistic insights, ethical regulation, and clinical validation. Multi-diseased patients constitute a challenge for EV-profile interpretation.

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
