# Peer review of "Extracellular Vesicles as Biomarkers in Chronic Hepatobiliary Diseases: An Overview of Their Interplay"

_ijms, 2025, doi:10.3390/ijms26136333_

Round 1
Reviewer 1 Report
Comments and Suggestions for Authors
The article, “Extracellular Vesicles as Biomarkers in Chronic Hepatobiliary Diseases: An Overview of their interplay,” is a comprehensive review of current research, but has several significant weaknesses:
1. While the article covers a wide range of topics from MASLD and ALD to cancer and autoimmune biliary diseases, it lacks in-depth analysis in key areas, e.g., numerous EV molecules are mentioned as biomarkers for MASLD and HCC, but there is no critique of their sensitivity, specificity, or clinical validation. Often used phrases such as “promising biomarker” are unsupported by meta-analysis or assessment of the level of evidence.
2.The article does not address the comparison of the diagnostic efficacy of EVs against classical biomarkers, e.g. ALT, AFP, CA19-9, and the compatibility of EVs with imaging modalities or with biopsy in clinical realities.
3 In the “Limitations” section, the authors point to the lack of standardization of EV isolation, the heterogeneity of molecules and the lack of validated clinical methods. However, the topics of cost, infrastructure and compatibility with current diagnostic systems are not elaborated on. In addition, there is no mention of potential obstacles to interpreting the results of EVs in multi-disease patients.
Author Response
The article, “Extracellular Vesicles as Biomarkers in Chronic Hepatobiliary Diseases: An Overview of their interplay,” is a comprehensive review of current research, but has several significant weaknesses:
While the article covers a wide range of topics from MASLD and ALD to cancer and autoimmune biliary diseases, it lacks in-depth analysis in key areas, e.g., numerous EV molecules are mentioned as biomarkers for MASLD and HCC, but there is no critique of their sensitivity, specificity, or clinical validation.- Often used phrases such as “promising biomarker” are unsupported by meta-analysis or assessment of the level of evidence.
Response: Thank you for pointing out this weakness. We revised our tables with AUC, sensitivity, specificity for those biomarkers with available ROC analysis. In response, we have now included additional analysis discussing the diagnostic performance (e.g., sensitivity, specificity, and AUC values) of specific EV-based biomarkers. For instance, we now highlight EV-miR-122 in MASLD (AUC: 0.77), and miR-451a and miR-1246 in gallbladder cancer (AUC: 0.664 and 0.646 respectively), as well as their combined performance with classical markers (e.g., CEA + CA19-9 + miR-1246: AUC = 0.816; Sensitivity = 72%; Specificity = 90.8%) We also expanded our tables (Table 2b, Table 3, Table 9) to include quantifiable metrics where available, allowing a more nuanced view of the biomarker efficacy. Although a meta-analysis was beyond the scope of this overview, we now provide clearer distinctions between EVs with preliminary data and those with stronger clinical validation. We acknowledge that no formal meta-analysis ( which was not the aim of this study, but a concise review of the current literature) was conducted or cited in this review due to the heterogeneity and limited size of many of the underlying studies. We now explicitly state this limitation and note that current literature is not yet sufficient in most cases to support a rigorous meta-analysis. However, we emphasize that where pooled data was available (e.g., AUC values from multiple studies), we have reported them accordingly.
2.The article does not address the comparison of the diagnostic efficacy of EVs against classical biomarkers, e.g. ALT, AFP, CA19-9, and the compatibility of EVs with imaging modalities or with biopsy in clinical realities.
Response:
This is an important point, and we have revised the manuscript to include direct comparisons where available. For example, in the context of MASLD, EV-miR-135a-3p showed a superior AUC (0.849) compared to ALT (AUC = 0.672). Similarly, for HCC, we discuss how EV-miR-21-5p combined with AFP yields improved diagnostic accuracy (AUC = 0.85; sensitivity = 95%) compared to AFP alone.
Furthermore, we now comment on the potential synergy between EVs and traditional modalities such as imaging and biopsy, particularly emphasizing the non-invasive, complementary role EVs can play in early disease detection and monitoring. This has been added to both the discussion and conclusions sections to better frame EVs as adjunct tools rather than standalone replacements at this stage of research.
Reviewer Comment 3: In the “Limitations” section, the authors point to the lack of standardization of EV isolation, the heterogeneity of molecules and the lack of validated clinical methods. However, the topics of cost, infrastructure and compatibility with current diagnostic systems are not elaborated on. In addition, there is no mention of potential obstacles to interpreting the results of EVs in multi-disease patients.
Response:
We appreciate the reviewer’s insightful suggestion. We have now expanded the “Limitations” section to include a detailed discussion regarding the issues that you pointed out.

Reviewer 2 Report
Comments and Suggestions for Authors
The review of the manuscript, "Extracellular Vesicles as Biomarkers in Chronic Hepatobiliary Diseases: An Overview of their interplay," received overwhelmingly positive feedback with a few suggestions for further enhancement.
Positive Comments:
- High Quality and Relevance: The manuscript is a well-structured, comprehensive, and highly relevant review that effectively addresses a rapidly evolving field.
- Excellent English Language: The English language level and scientific writing quality are consistently excellent throughout the text.
- Clarity and Cohesion: The review is clear, concise, and logically organized with effective transitions, making complex information accessible.
- Thoroughness: It provides a thorough overview of EV biology and their interplay in various chronic hepatobiliary diseases, supported by an extensive and up-to-date reference list.
Suggestions for Improvement:
- Deeper Dive into Specific Areas: Consider highlighting emerging areas, controversies, or challenges in clinical translation, such as the importance of standardized methodologies for EV isolation and characterization.
- Enhanced Future Directions: Expanding on the future outlook, particularly regarding the hurdles and opportunities for translating EV biomarkers into routine clinical practice, would add valuable depth.
- Potential for Targeted Therapies: Briefly exploring how EV research might contribute to targeted therapies, beyond their biomarker role, could enrich the discussion.
In summary, the manuscript is of high quality and makes a significant contribution to the field, with minor suggestions aimed at further strengthening its impact and comprehensiveness.
Author Response
Reviewer 2:
General comments
The review of the manuscript, "Extracellular Vesicles as Biomarkers in Chronic Hepatobiliary Diseases: An Overview of their interplay," received overwhelmingly positive feedback with a few suggestions for further enhancement.
Positive Comments:
- High Quality and Relevance: The manuscript is a well-structured, comprehensive, and highly relevant review that effectively addresses a rapidly evolving field.
- Excellent English Language: The English language level and scientific writing quality are consistently excellent throughout the text.
- Clarity and Cohesion: The review is clear, concise, and logically organized with effective transitions, making complex information accessible.
- Thoroughness: It provides a thorough overview of EV biology and their interplay in various chronic hepatobiliary diseases, supported by an extensive and up-to-date reference list.
Response: Thank you for your valuable comments. We sincerely appreciate the reviewer’s positive assessment and constructive comments.
Suggestions for Improvement:
- Deeper Dive into Specific Areas: Consider highlighting emerging areas, controversies, or challenges in clinical translation, such as the importance of standardized methodologies for EV isolation and characterization.
- Enhanced Future Directions: Expanding on the future outlook, particularly regarding the hurdles and opportunities for translating EV biomarkers into routine clinical practice, would add valuable depth.
- Potential for Targeted Therapies: Briefly exploring how EV research might contribute to targeted therapies, beyond their biomarker role, could enrich the discussion.
In summary, the manuscript is of high quality and makes a significant contribution to the field, with minor suggestions aimed at further strengthening its impact and comprehensiveness.
Response: We sincerely appreciate the reviewer’s constructive comments. The suggested refinements have been carefully incorporated to strengthen the manuscript’s scientific depth, translational relevance, and future outlook.
More particularly, we expanded the limitations with:
- Current isolation methods are related to re-sources with implying several limitations such as the cost and infrastructure, with the standardization of these being necessary for clinical laboratories.
- Integration of EV analysis in combination of existing diagnostic modalities such as biopsy, serological lab tests and imaging modalities that will provide a multi-modal interpretation, is considered a challenge as it requires algorithmic approach and harmonization.
- On the top of that the common EV profiling in several diseases constitute another challenge for the interpretation of EV finding in the presence of multiple diseases in each patient (ex. chronic hepatitis B and MASLD), so the development of disease-specific panels may overcome this issue.
- Finally, inconsistencies in EV definitions, characterization, and reporting across studies hinder reproducibility and consensus.
Thank you for underlying the addition of EV research in targeted therapies, however our scope was to underline their role as biomarkers. We added just few words regarding this issue, however we cannot expand it , as the whole manuscript is focused on their diagnostic potential. We also enhanced the paragraph of conclusions & future directions.
“Finally, in addition to EV’s role as diagnostic and prognostic biomarkers, they have an emerging role as promising tools for targeted therapies, due to their ability to transfer several cargoes. This ability opens new horizons for future perspectives in EV research and engineering for drug delivery and personalized hepatobiliary disease management.”

Round 2
Reviewer 1 Report
Comments and Suggestions for Authors
Thank you to the authors for their diligent review and for incorporating my suggestions. I believe that in its current form the article should be published.
Author Response
Τhank you for substantial review report that increased the quality of our manuscript